# Artificial Intelligence and Its Impact on the Management of Lumbar Degenerative Pathology: A Narrative Review

**DOI:** 10.3390/medicina61081400

**Published:** 2025-08-01

**Authors:** Alessandro Trento, Salvatore Rapisarda, Nicola Bresolin, Andrea Valenti, Enrico Giordan

**Affiliations:** 1Department of Neuroscience, University of Verona, 37126 Verona, Italy; alessandro.trento@aulss2.veneto.it; 2Department of Neuroscience, University of Padua, 35128 Padua, Italy; salvatore.rapisarda@aulss2.veneto.it (S.R.); nicola.bresolin@aulss2.veneto.it (N.B.); andrea.valenti@aulss2.veneto.it (A.V.); 3Neurosurgical Department, Aulss2 Marca Trevigiana, 31100 Treviso, Italy

**Keywords:** artificial intelligence, AI, lumbar, surgery, degenerative, spine

## Abstract

In this narrative review, we explore the role of artificial intelligence (AI) in managing lumbar degenerative conditions, a topic that has recently garnered significant interest. The use of AI-based solutions in spine surgery is particularly appealing due to its potential applications in preoperative planning and outcome prediction. This study aims to clarify the impact of artificial intelligence models on the diagnosis and prognosis of common types of degenerative conditions: lumbar disc herniation, spinal stenosis, and eventually spinal fusion. Additionally, the study seeks to identify predictive factors for lumbar fusion surgery based on a review of the literature from the past 10 years. From the literature search, 96 articles were examined. The literature on this topic appears to be consistent, describing various models that show promising results, particularly in predicting outcomes. However, most studies adopt a retrospective approach and often lack detailed information about imaging features, intraoperative findings, and postoperative functional metrics. Additionally, the predictive performance of these models varies significantly, and few studies include external validation. The application of artificial intelligence in treating degenerative spine conditions, while valid and promising, is still in a developmental phase. However, over the last decade, there has been an exponential growth in studies related to this subject, which is beginning to pave the way for its systematic use in clinical practice.

## 1. Introduction

Degenerative conditions of the lumbar spine are among the leading causes of pain and disability in the adult population. Approximately 266 million people (3.6% of the global population) suffer from low back pain associated with arthritic changes in the lumbar spine [1]. These conditions negatively affect quality of life by causing significant functional limitations in daily activities. They primarily include degenerative disc disease, spinal canal stenosis, and herniated discs. They are often associated with radiculopathy, back pain, loss of function, and reduction in lumbar lordosis.

Nonoperative strategies, such as percutaneous steroid injections and physical therapy, are typically effective as first-line treatments and are often combined with physiotherapy. In cases of persistent pain or neurological deficits, surgical interventions—such as discectomy, laminectomy, or spinal fusion—are preferred and considered definitive.

In recent years, applications of artificial intelligence (AI) have expanded rapidly in the healthcare sector, offering the potential to revolutionize the field by enhancing diagnostic accuracy and predicting clinical outcomes [2,3]. This trend extends to spine-related pathologies, both in diagnosis and treatment. The objective of this review is to collect, analyze, and highlight the role of AI in managing degenerative lumbar spine conditions, with a focus on the diagnosis and prognosis of lumbar disc herniation and spinal stenosis, as well as predictive factors for lumbar fusion surgery.

## 2. Results and Discussion

### 2.1. Study Characteristics

The PubMed, Scopus, and Web of Science databases were searched for articles published from May 2015 to May 2025 using the following keywords: “lumbar disc herniation AND artificial intelligence”, “lumbar stenosis AND artificial intelligence”, and “lumbar fusion surgery AND artificial intelligence”. Eligible studies included English-language articles.

Four authors (N.B., A.V., S.R., and A.T.) conducted the bibliographic and review search. A total of 331 articles were initially identified. Following a screening process based on titles and abstracts and subsequent full-text evaluation, 96 studies were deemed eligible for inclusion.

A summary of AI’s evolution over the past decade, based on our research, is shown in Figure 1.

To avoid confusion, the authors defined several terms related to the broad umbrella of artificial intelligence (AI) prior to conducting the search.

Machine learning (ML) is a subset of AI that focuses on optimization. When implemented correctly, it enables predictions that minimize errors compared to simple guessing. For example, companies use ML to recommend products to customers based on their prior browsing and purchasing behavior.

Deep learning (DL), a further subset of ML, differs primarily in how it learns and the volume of data it requires. DL automates much of the feature extraction process, reducing the need for manual human input. It also leverages large datasets, enhancing its predictive capabilities.

To effectively compare different predictive models, it is essential to understand that the area under the curve (AUC) serves as a key metric for evaluating model performance. AUC quantifies how well an AI model distinguishes between patients with specific characteristics and those without, using a probabilistic framework. An AUC score between 0.7 and 0.8 indicates acceptable performance; scores from 0.8 to 0.9 are considered good; and scores above 0.9 are excellent. In contrast, scores near 0.5 suggest performance equivalent to random chance.

### 2.2. AI for Lumbar Spinal Stenosis

Lumbar spinal stenosis (LSS) is one of the most common spinal conditions affecting adults, particularly the elderly. The clinical presentation typically includes low back pain radiating to the lower extremities, often accompanied by numbness. Symptoms usually worsen with walking, leading to significant limitations in daily activities [4].

#### 2.2.1. Diagnosis

In recent years, ML and DL models have been extensively developed for the diagnosis and classification of lumbar spinal stenosis. Most studies have focused on magnetic resonance imaging (MRI), the gold-standard diagnostic modality due to its detailed assessment of the central canal, lateral recesses, and foramina, along with its excellent soft tissue contrast [5].

Although MRI provides valuable diagnostic information, executing and interpreting scans is time-consuming and heavily dependent on the radiologist’s expertise. Currently, no automatic quantitative criteria exist for diagnosing lumbar spinal stenosis [6]. Therefore, automated grading systems are warranted to reduce radiologists’ workloads and improve diagnostic accuracy.

One of the first AI models, developed by Jamaludin et al. [7], used a multitask architecture to classify degenerative conditions of the lumbar spine, including central canal stenosis. The condition was assessed in a binary fashion (present or absent), and only sagittal images were used.

Another notable early development was Deep Spine, created by Lu and colleagues [8], which evaluated and classified central canal and foraminal stenosis using both axial and sagittal images. This system was based on a weakly supervised interpretation of radiology reports. More recently, Hallinan et al. [9] developed a DL model to automatically detect and classify central canal, lateral recess, and neural foramina stenosis. Their study involved 446 patients and employed 12,403 axial T2-weighted images along with 6161 sagittal T1-weighted images for training, validation, and testing. The model showed strong agreement with radiologists in the dichotomous classification of central canal and lateral recess stenosis (normal or mild vs. moderate or severe). Nonetheless, the level of agreement was slightly lower for neural foraminal stenosis.

However, these studies did not provide quantitative measurements for stenosis classification. A model developed by Bharadwaj and colleagues [10] aimed to address this limitation by integrating a decision tree classifier that used quantitative measurements of the cross-sectional areas of the dural sac and intervertebral discs. This model also assessed facet arthropathy, an important contributor to lower back pain. While the binary classification of central canal stenosis, neural foramina stenosis, and facet arthropathy demonstrated accuracy, the study had notable limitations, including a small sample size (200 patients) and the lack of an external model of validation.

Furthermore, Van der Graaf et al. [11] developed an AI-based model for classifying central canal stenosis that automatically extracted the cross-sectional area, anteroposterior diameter of the dural sac, and cerebrospinal fluid (CSF) signal loss [12]. Their algorithm, based on the Lee classification [13] and using only sagittal images, achieved a sensitivity of 93% and a specificity of 91%. These results were comparable to assessments made by two expert radiologists.

Computed tomography (CT) imaging is generally not preferred for evaluating lumbar spinal stenosis due to increased noise from the bony structures surrounding the spinal canal. However, CT provides better delineation of the ligamentum flavum compared to MRI [14]. Based on this observation, Miyo et al. [15] applied a deep learning reconstruction method to 30 lumbar CT scans and compared the results to those obtained using hybrid iterative reconstruction, an older technique for enhancing CT image quality. The authors reported improved quantitative image noise and better interobserver agreement regarding the degree of lumbar spinal stenosis.

Artificial intelligence solutions can also be applied to simpler diagnostic tools. For example, Kim et al. [16] developed a deep learning algorithm to diagnose central lumbar spinal stenosis using radiographs. The study included 2303 patients with severe central canal stenosis confirmed by MRI and 2341 controls. Lateral lumbar radiographs in neutral, flexion, and extension positions—comprising 6325 images in the stenosis group and 6117 in the control group—were analyzed for disc height, intervertebral foramen height, pedicle length, and facet joint hypertrophy. Among the models trained, one achieved an area under the ROC curve (AUC) of 90%, with an accuracy of 81.8%, sensitivity of 85.9%, and specificity of 77.8% in the neutral position. These findings suggest that artificial intelligence may enable the use of simple and cost-effective diagnostic tools, such as a radiograph, to identify lumbar spinal stenosis.

It is important to note that the clinical presentation of lumbar spinal stenosis may not always align with radiological findings [17]. Therefore, self-reported questionnaires have been validated to assist in the symptomatic diagnosis of lumbar spinal stenosis [18]. Abel and colleagues [19] developed several ML models to identify lumbar spinal stenosis based on a 26-question survey that assessed pain severity and type, activities limited by pain, motor impairment, and overall physical and mental health. The best-performing model achieved an area under the curve (AUC) of 96%, with a sensitivity of 94% and a specificity of 88% in classifying patients with or without lumbar spinal stenosis. These results demonstrate the potential of AI to support diagnosis using a simple clinical assessment tool.

#### 2.2.2. Treatment

Machine learning algorithms can also enhance the surgical management of lumbar spinal stenosis by facilitating the preoperative identification of patients who may benefit from surgery. In 2019, Siccoli et al. [20] developed a model based on 15 variables, including outcome measures such as the Numeric Rating Scale (NRS) for back and leg pain and the Oswestry Disability Index (ODI), as well as demographic characteristics such as age, sex, and body mass index (BMI). These data were extracted from 635 patients who underwent lumbar decompression. Clinical success was defined as achieving the minimum clinically important difference (MCID), characterized by an improvement in ODI or NRS of ≥30%. The model demonstrated the feasibility of predicting MCID at both 6 weeks and 12 months postoperatively, with prediction accuracies for NRS and ODI ranging from 62% to 85% and AUC values as high as 0.92 (for back pain NRS at 6 weeks).

More recently, Wilson and colleagues [21] developed an ML model to predict the need for surgery using axial T2-weighted MRI scans. Their study included 80 patients who underwent decompression surgery and 60 controls. As a measure of spinal stenosis, the authors considered the maximum percentage reduction in the spinal canal area on lumbar MRI, corresponding to the most compressed level. The model demonstrated high predictive accuracy, with an AUC greater than 0.88, in identifying patients who would undergo subsequent spinal decompression.

One potential drawback of current machine learning approaches is the high discrepancy between clinical symptoms and radiological findings of degenerative changes [22]. To address this limitation and predict the need for lumbar decompression based on both clinical and radiological data, Mourad et al. [23] developed a novel hybrid model using 500 medical vignettes. Each vignette included 36 variables representing clinical symptoms, MRI features, and demographic factors. The model was constructed using a weighted average between a Bayesian network, which incorporated expert opinion, and a machine learning model trained on the same 36-variable dataset. In a dichotomous classification framework (weak recommendation vs. strong recommendation for surgery), the hybrid model outperformed the recommendations of five individual experts, achieving an AUC of 0.92 compared to 0.84. The authors also developed a separate ML model using the same dataset, which achieved comparable accuracy [24].

#### 2.2.3. Prognosis

AI models have applications in the postoperative care of patients undergoing lumbar decompression surgery. Patients often experience prolonged hospital stays due to difficulties with mobilization, which can negatively impact both healthcare costs and patient autonomy [25]. Addressing this issue, Ogink et al. [26] evaluated four ML algorithms to predict discharge destinations—either rehabilitation or nursing facilities—after lumbar decompression. Their dataset included 28,600 patients from the American College of Surgeons National Surgical Quality Improvement Program. The variables analyzed were age, body mass index (BMI), American Society of Anesthesiologists (ASA) classification, functional status, number of surgical levels, fusion, diabetes, preoperative hematocrit, and serum creatinine. The neural network model demonstrated promising accuracy (AUC = 0.75) in distinguishing patients discharged home from those requiring rehabilitation. Similarly, Saravi and colleagues [27] applied various algorithms to a retrospective cohort, considering clinical, demographic, and surgical variables such as sex, age, BMI, operation time, and C-reactive protein (CRP) levels, to predict hospital length of stay after lumbar decompression. One of the deep learning algorithms they tested achieved excellent accuracy (AUC = 0.87) in predicting prolonged hospital stays.

### 2.3. AI for Lumbar Disc Herniation

Lumbar disc herniation with radiculopathy is a common and debilitating spinal disorder that requires a multidisciplinary approach, in which artificial intelligence (AI) shows significant potential.

#### 2.3.1. Diagnosis

Several DL models have demonstrated high performance in identifying and classifying disc pathologies using MRI and computed tomography (CT), thereby reducing diagnostic subjectivity [28,29,30,31,32]. Notably, Xu et al. [33] developed a DL model capable of simultaneously localizing and classifying lumbar disc herniation. The model analyzed axial T2-weighted MR sequences from 1496 patients, and its performance was compared with that of three spinal surgeons. The algorithm achieved diagnostic performance comparable to that of the experts, demonstrating reasonable accuracy (sensitivity: 87.0% for grading and 84.0% for localization; specificity: 95.5–94.4%, respectively). Despite its promise, the model’s performance declined significantly during external testing.

However, surgical selection for lumbar disc herniation relies not solely on imaging findings but primarily on clinical–radiological correlation. Staartjes et al. [34] developed a machine learning classification system that correlates radiological features with specific pain generators in lumbar degenerative pathology. Their study included 262 surgical candidates diagnosed with lumbar disc herniation, lumbar spinal stenosis, or discogenic chronic low back pain, all of whom underwent the five-repetition sit-to-stand test (5R-STS). By incorporating the test results alongside the patient’s age, gender, height, and weight, the model distinguished among the three pathologies with an accuracy of 96.2%, demonstrating excellent discrimination. Nevertheless, the model’s clinical applicability remains limited due to a lack of external validation.

#### 2.3.2. Treatment

Beyond diagnosis, AI technologies have also demonstrated significant potential to enhance preoperative planning, intraoperative support, and procedural precision in spinal surgery. Zhu et al. [35] developed an AI-based three-dimensional (3D) model of the lumbosacral region to improve personalized trajectory planning for percutaneous lumbar endoscopic discectomy at the L5–S1 level. This reconstruction achieved accuracy comparable to that of a manual-segmentation-based 3D model in depicting the L5 and S1 vertebrae, the L5–S1 disc, the lumbosacral nerve roots, the iliac bone, and the skin. This region is of particular interest due to the anatomical challenge posed by the iliac crest [36].

Similarly, Yamada and colleagues [37] applied an AI-enhanced MRI 3D model to identify patients with an L5–S1 disc herniation suitable for endoscopic transforaminal removal. Among the 52 cases analyzed, 13 were deemed operable. All endoscopic surgeries proceeded without complications and resulted in positive outcomes, notably pain reduction.

AI solutions can also be applied to intraoperative imaging. Cui et al. [38] collected 65 videos of endoscopic transforaminal discectomy, extracting over 10,000 images. They then trained an ML detection system to identify the dura mater and nerve roots from this dataset. The algorithm demonstrated excellent performance, achieving a sensitivity of 90.9%, specificity of 93.7%, and accuracy of 92.3%. This performance matched that of an expert spinal endoscopist and significantly exceeded that of less experienced surgeons.

#### 2.3.3. Prognosis

AI algorithms show promise in predicting surgical outcomes. ML models have demonstrated high accuracy in forecasting patient-specific results, thereby enabling more personalized treatment planning. Yen et al. [39] developed a model to predict prolonged opioid use following lumbar discectomy based on data from 1316 patients. This model achieved acceptable accuracy with an area under the curve (AUC) of 0.76. Similarly, Wang et al. [40] used a DL MRI model combined with clinical features to predict one-year outcomes after tubular microdiscectomy. This study involved 548 patients and incorporated sagittal and axial T2 sequences alongside preoperative clinical characteristics. One of the tested DL models yielded optimal results, with an AUC of 0.86 for internal validation and 0.83 for external validation.

### 2.4. AI for Lumbar Fusion Surgery

Lumbar spinal fusion has become one of the most commonly performed procedures in modern spine surgery. However, the rapid increase in procedure volume has been accompanied by a growing burden of perioperative complications and hospital readmissions, particularly among aging populations with complex comorbidities [41,42,43]. As healthcare systems shift toward value-based care, stratifying risk and personalizing perioperative management have become essential. Artificial intelligence (AI), particularly machine learning (ML) techniques, is increasingly recognized as a promising tool for addressing these challenges. Traditional statistical methods, such as logistic regression (LR), have played a crucial role in identifying key perioperative risk factors for lumbar fusion [44,45,46,47,48,49]. However, LR’s assumptions of linearity and limited ability to handle complex interactions have driven the adoption of more sophisticated ML models [50,51]. ML offers advantages in processing large datasets and detecting non-linear relationships. Several studies have demonstrated that ML models frequently outperform LR models in predictive accuracy for surgical outcomes [45,46,47,48].

#### 2.4.1. Outcome Prediction

ML approaches have been explored across a wide spectrum of spinal pathologies [44,49,52,53,54]. Although several studies have shown promise, their scope, design, and methodology vary considerably. Berjano et al. [55] developed an ML model to predict favorable outcomes after lumbar arthrodesis. The study included 1243 patients and considered demographic, clinical, and surgical features. A good outcome at six-month follow-up was defined as an improvement of more than 12.7 points on the Oswestry Disability Index (ODI) [56]. The model demonstrated promising results, with a sensitivity of 74.3%, specificity of 79.4%, and accuracy of 75.8%. However, the algorithm lacked external validation, and the surgical approach was not specified.

Schönnagel et al. [57] employed XGBoost, a type of ML algorithm, to predict persistent lower back pain after lumbar fusion in patients with degenerative spondylolisthesis, achieving an AUC of 0.81. Nevertheless, the single-center design and small sample size (135 patients) raised concerns about overfitting and limited generalizability [58,59]. In contrast, Fatima et al. [60] applied ML to a larger dataset comprising over 80,000 patients with lumbar degenerative spondylolisthesis to predict 30-day adverse events. Their LR-based model achieved an AUC of 0.70 and outperformed traditional frailty indices [61]. They also developed a web-based calculator to facilitate clinical translation.

#### 2.4.2. Complication Prediction

Several recent studies have sought to leverage machine learning’s capabilities to predict complications, readmissions, discharge destinations, and persistent pain in patients undergoing lumbar fusion. Dong et al. [62] utilized a support vector machine to predict poor outcomes following lumbar interbody fusion, demonstrating how ML can incorporate imaging-derived variables. Their results highlighted the predictive value of parameters, such as disc height and spinal alignment, which closely correlate with clinical outcomes [63,64,65,66,67,68,69].

Bui et al. [70] developed an ML pipeline to predict interbody cage height and the degree of pelvic mismatch after L4–L5 transforaminal lumbar interbody fusion (TLIF) surgery using preoperative X-ray images. The automated pipeline consisted of two stages: first, a DL model extracted essential features from the X-ray images; second, five ML algorithms were trained to identify optimal models for predicting interbody cage height and postoperative pelvic mismatch. Although the accuracy of the tested models was moderate, this study represents an important initial step toward developing tools to predict changes in sagittal balance following interbody fusion surgery.

Shah et al. [44] developed ML models using a national discharge database of nearly 39,000 patients to predict major complications and readmission risk. Their models outperformed LR and identified novel risk factors. These insights highlight modifiable factors, such as diabetes and cardiovascular disease, which may benefit from optimization prior to surgery, thereby helping clinicians provide more effective patient counseling. Janssen et al. [71] incorporated preoperative physical tests into their ML framework and identified aerobic capacity, muscle endurance, and flexibility as strong predictors of recovery. Their findings suggest that functional metrics—often overlooked in large datasets—hold significant clinical value. This perspective aligns with the existing literature emphasizing the importance of prehabilitation and physical conditioning in improving surgical outcomes [72,73,74,75,76].

Fusion surgery can be challenging, with prolonged operative time representing a significant risk factor for complications, such as infections [77]. Li and colleagues [78] recently developed a machine learning model to predict extended operation times in patients undergoing posterior lumbar interbody fusion. Their study included 3233 patients from 22 Chinese institutions. The model incorporated demographic variables, perioperative details, and laboratory results, achieving good predictive performance with an AUC of 0.82 in identifying patients at high risk of prolonged surgery (>240 min).

Xiong et al. [79] developed a model to predict postoperative infection by evaluating 584 patients who underwent fusion surgery. Their model achieved an AUC of 0.87 by considering preoperative variables, such as albumin level, diabetes status, intraoperative dural tears, and history of rheumatic disease.

AI has also been applied to predict cage subsidence [80]. Zou et al. [81] proposed a predictive model based on data from 305 patients across three centers undergoing lumbar fusion surgery. By integrating radiological features from CT and MR scans with clinical variables—including sex, age, number of surgical segments fused, body mass index (BMI), and presence of osteoporosis—the model achieved high accuracy, with AUC values up to 0.94. This model holds promise for improving clinical decision-making and reducing the need for revision surgeries.

The application of deep learning (DL) in lumbar fusion surgery presents both opportunities and challenges. Kuris et al. [82] applied an artificial neural network (ANN) to predict readmissions after various types of lumbar fusion in over 63,000 patients. Although the model achieved high classification accuracy (~94%), its AUC values were modest (0.64–0.65), indicating limited discriminative ability and underscoring the importance of selecting appropriate performance metrics [83,84]. Similarly, Hopkins et al. [49] demonstrated that DL neural networks can outperform traditional models in predicting 30-day readmission following lumbar fusion. However, their study revealed that strong predictions often depend on intra- and postoperative variables, limiting the model’s utility for early risk stratification. This work highlights both the potential and constraints of deep neural networks (DNNs) in informing hospital policy and emphasizes the need for AI systems that integrate data across the entire perioperative timeline [72,75,85]. Kim et al. [86] showed that ANN models outperformed LR in predicting complications—including mortality and wound issues—using data from over 22,000 posterior lumbar fusion cases. These findings align with those of Dreiseitl and Ohno-Machado [87], reinforcing the expanding role of AI in clinical prognostication and its potential for implementation in practice.

To address the challenge of AI model interpretability, some researchers have adopted explainable AI techniques. Guo et al. [88] developed a model to predict cerebrospinal fluid leakage in 3505 lumbar fusion cases, achieving an AUC of 0.87. This model not only demonstrated high accuracy but also highlighted clinically relevant imaging and clinical variables—such as ligamentum flavum thickness and facet joint degeneration—that are both measurable and potentially modifiable.

#### 2.4.3. Cost Prediction

Other researchers have focused on discharge planning and resource allocation. Jain et al. [89] applied various ML models to predict discharge to a facility, 90-day readmission, and medical complications in nearly 38,000 patients undergoing long-segment lumbar fusion [90,91]. Similarly, Cabrera et al. [92] stratified ML models by age group to predict non-home discharge across 39,254 patients, enabling more precise perioperative planning compared to static models. These studies suggest that future AI applications must balance scalability with the inclusion of richer, patient-centered features to support individualized care planning [93,94,95].

Several investigations have also addressed cost and policy implications. Karnuta et al. [96] developed a classifier to predict inpatient cost, length of stay, and discharge disposition in over 38,000 non-scoliosis lumbar fusion patients. Achieving AUC values exceeding 0.88, the model facilitated the development of patient-specific payment models. This represents a critical step toward more equitable reimbursement structures that account for patient complexity—an increasingly important consideration within bundled-payment environments [97,98,99].

### 2.5. Limitations

The primary limitation of this review is its narrative nature in describing the AI landscape within common degenerative lumbar spine conditions. The study search did not follow a predefined protocol, and the data were neither systematically abstracted nor quantitatively analyzed. Instead, the findings were reported descriptively by the authors. Nonetheless, this review aims to capture and illustrate the current potential of AI and its deep learning models as they relate to the most prevalent degenerative lumbar spine pathologies.

## 3. Conclusions

The application of artificial intelligence (AI) and machine learning (ML) in the treatment of spinal pathologies and surgical procedures is advancing rapidly. Numerous models show strong potential to enhance diagnostic accuracy, optimize surgical planning, and predict adverse events, readmissions, recovery times, and associated costs. The most successful models are typically those trained on large, well-structured datasets and enriched with clinically relevant modifiable variables. Despite these promising results, the majority of current studies rely on retrospective data, often derived from administrative records that lack critical intraoperative variables, postoperative functional assessments, and detailed imaging data. Moreover, model performance varies considerably depending on cohort size, event rates, feature engineering, and validation methodologies. Few studies report external validation, raising concerns that many high-performing models may be overfitted to specific datasets or institutions.

To realize the full potential of AI in lumbar degenerative pathology management, models must be interpretable, transparent, and seamlessly integrated into electronic health records to facilitate tailored healthcare delivery. Additionally, ethical considerations—including data privacy, biases within training datasets, and risks of overreliance on automation—require careful attention. Future research is essential to ensure that AI becomes a safe, effective, and trustworthy tool in the management of lumbar degenerative pathology.

## Figures and Tables

**Figure 1 medicina-61-01400-f001:**
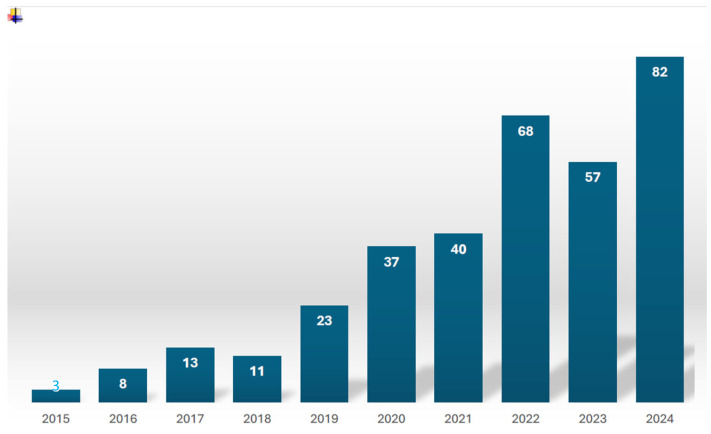
The annual number of articles related to lumbar degenerative pathology and artificial intelligence.

## Data Availability

Data are contained within the article.

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
