# Peer review of "Artificial Intelligence and Its Impact on the Management of Lumbar Degenerative Pathology: A Narrative Review"

_medicina, 2025, doi:10.3390/medicina61081400_

Round 1

Reviewer 1 Report

Comments and Suggestions for Authors

The authors describe their manuscript as a narrative review, yet the methodology is presented in the style of a systematic review. They must decide whether to proceed as a narrative or a systematic review.

If they choose a systematic review, the manuscript will require extensive revision before minor edits can be addressed in a second round. Specifically, the Materials and Methods section should be restructured into standard systematic‐review subheadings (e.g., Search Strategy, Inclusion Criteria, Study Selection, Data Extraction, Quality Assessment). I recommend consulting a published example—such as this MDPI article (https://www.mdpi.com/2072-6694/16/11/2089)—to guide the organization of that section. The Results and Discussion should be separated into distinct sections; the review should be registered in an appropriate public registry (e.g., OSF or PROSPERO); and the included studies and their findings should be summarized in tables.

If the authors opt for a narrative review, the detailed Materials and Methods section becomes redundant. In that case, they should combine Results and Discussion into clearly labeled subsections to improve readability and manuscript flow.

Once the authors have made their choice and revised accordingly, I would be happy to conduct a second‐round review.

Author Response

Comment: If the authors opt for a narrative review, the detailed Materials and Methods section becomes redundant. In that case, they should combine Results and Discussion into clearly labeled subsections to improve readability and manuscript flow.

We sincerely thank the reviewers for their valuable insights and for highlighting this issue. We acknowledge the concern and apologize for any confusion caused. As a result, we have revised the Materials and Methods section to summarize the selection process and have redefined the PRISMA flow diagram, simplifying it to present a clearer study selection diagram (Figure 1). We have labeled the subparagraphs for improved readability. The sections on "AI for Lumbar Spinal Stenosis" and "AI for Lumbar Disk Herniation" are divided into paragraphs covering diagnosis, treatment, and prognosis (page 3, line 88; page 4, line 157; page 5, line 189 and page 5, line 212; page 6, line 233; page 6, line 254 respectively). Meanwhile, the "AI for Lumbar Fusion Surgery" section is organized into paragraphs discussing outcomes, complications, and costs. (page 7, line 282; page 7, line 301; page 8, line 370)

Reviewer 2 Report

Comments and Suggestions for Authors

  1. Methodological concerns and scientific rigor
    • The manuscript lacks systematic review methodology. Literature search strategy should follow PRISMA guidelines with detailed documentation
    • Inclusion/exclusion criteria need clearer definition (currently only mentions English language and timeframe)
    • Quality assessment (e.g., QUADAS-2) for included studies is missing
    • The narrative review format, while acknowledged, does not meet current standards for evidence synthesis
  2. Limited novelty and clinical impact
    • The manuscript primarily summarizes existing studies without providing new analysis or unique insights
    • No meta-analysis or statistical synthesis of AI performance metrics across studies
    • Lacks concrete recommendations for clinical implementation
    • Missing discussion of cost-effectiveness and regulatory pathways for AI adoption
    • Insufficient attention to implementation barriers and ethical considerations
  3. Results presentation and analysis
    • Need standardized comparison tables for AI performance metrics across different pathologies
    • Should include comprehensive metrics beyond AUC (sensitivity, specificity, PPV, NPV)
    • External validation status of each study must be clearly indicated
    • Critical comparison of different ML/DL architectures and their specific advantages is missing
  4. Clinical significance underexplored
    • No discussion of minimum dataset requirements for clinical implementation
    • Lacks analysis of which clinical scenarios benefit most from AI integration
    • Missing discussion of liability issues and regulatory approval processes
    • Insufficient attention to the gap between research findings and real-world application

Minor Issues:

  1. Figure and table improvements
    • Figure 2 needs specific numerical values added to the trend graph
    • A comprehensive comparison table summarizing AI studies by pathology would enhance readability
    • Consider adding a table comparing traditional vs AI-assisted diagnostic accuracy
  2. Literature completeness
    • Limited representation of Asian studies, particularly relevant given the submission to an international journal
    • Consider adding recent 2025 publications on AI in endoscopic spine surgery
  3. Structural enhancements
    • Add brief summaries at the end of each major section
    • Include a practical framework or flowchart for clinical implementation
Comments on the Quality of English Language

    • Critical error on page 4, line 113: "weekly supervised interpretation" should be "weakly supervised interpretation" (this fundamentally changes the meaning)
    • Inconsistent reference formatting throughout ([number] vs number format)
    • Missing reference numbers in multiple locations (e.g., page 10, line 318: "Janssen et al.71" should be "Janssen et al.[71]")
    • Spacing errors and formatting inconsistencies that detract from professional presentation

Author Response

Comment 1: Methodological concerns and scientific rigor

    • The manuscript lacks systematic review methodology. Literature search strategy should follow PRISMA guidelines with detailed documentation
    • Inclusion/exclusion criteria need clearer definition (currently only mentions English language and timeframe)
    • Quality assessment (e.g., QUADAS-2) for included studies is missing
    • The narrative review format, while acknowledged, does not meet current standards for evidence synthesis

We would like to thank the reviewers for their valuable insights and appreciate the suggestions provided. Our paper is structured as a narrative review to provide a comprehensive overview of the relatively new applications of artificial intelligence in spinal surgery from the perspective of a spine surgeon. Therefore, we intentionally chose not to conduct a systematic review and did not focus on technical details such as architectures or ethical considerations. We conducted the study search following the same rules and guidelines reserved for systematic reviews (i.e., adhering to PRISMA guidelines) to ensure compliance with the "a priori" search protocol and enhance the overall quality of the included studies.

 Comment 2: Limited novelty and clinical impact

    • The manuscript primarily summarizes existing studies without providing new analysis or unique insights
    • No meta-analysis or statistical synthesis of AI performance metrics across studies
    • Lacks concrete recommendations for clinical implementation
    • Missing discussion of cost-effectiveness and regulatory pathways for AI adoption
    • Insufficient attention to implementation barriers and ethical considerations

We did not address regulatory pathways or cost-effectiveness, as the examples we presented are research findings that are not routinely applied in clinical practice.

Comments on the quality of English language

  • Critical error on page 4, line 113: "weekly supervised interpretation" should be "weakly supervised interpretation" (this fundamentally changes the meaning)
  • Inconsistent reference formatting throughout ([number] vs number format)
  • Missing reference numbers in multiple locations (e.g., page 10, line 318: "Janssen et al.71" should be "Janssen et al.[71]")
  • Spacing errors and formatting inconsistencies that detract from professional presentation

We have corrected the critical error you pointed out (and fixed the reference formatting. (page 3, paragraph AI for lumbar spinal stenosis - diagnosis, line 105).

Round 2

Reviewer 1 Report

Comments and Suggestions for Authors

Authors did not adequately address the suggestions from the previous round of review.

I suggest that they re-read the review from first round and respond point-by-point with track changes in revised manuscript.

Author Response

We would like to express our sincere gratitude to the reviewers for their valuable insights, and we apologize for any delays that may have occurred. We have revised the manuscript by merging the discussion and results sections for clarity. To further enhance reader understanding, we have mantained an introductory section that outlines our research process. Additionally, we have labeled the subparagraphs to improve readability. The sections are "Study characteristics" (page 2, line 51); "AI for lumbar spinal stenosis" (page 3, line 85) with the subsections "Diagnosis" (page 3, line 91), "Treatment" (page 4, line 160), "Prognosis" (page 5, line 192); "AI for lumbar disc herniation" (page 5, line 210) with the subsections "Diagnosis" (page 5, line 215), "Treatment" (page 5, line 236), "Prognosis" (page 6, line 257); "AI for lumbar fusion surgery" (page 6, line 269) with the subsections "Outcome predicition" (page 6, line 285), "Complications prediction" (page 7, line 304), "Costs predicition" (page 8, line 373); "Limitations" (page 8, line 390).